# Magnetic Behavior of Carbon Materials Made from Biomass by Fe-Assisted Hydrothermal Carbonization

**DOI:** 10.3390/molecules24213996

**Published:** 2019-11-05

**Authors:** Mara Olivares, Silvia Román, Beatriz Ledesma, Alfredo Álvarez

**Affiliations:** 1Department of Mechanical, Energetics and Materials Engineering, Mérida University Center, 06800 Mérida, Spain; maraom@unex.es; 2Department of Applied Physics, Industrial Engineering School, University of Extremadura, Av. Elvas s/n, 06006 Badajoz, Spain; beatrizlc@unex.es; 3Department of Electric, Electronic and Automatic Engineering, Industrial Engineering School, University of Extremadura, Av. Elvas s/n, 06006 Badajoz, Spain; aalvarez@unex.es

**Keywords:** hydrocarbonization, pyrolysis, magnetic materials, adsorption

## Abstract

Biomass magnetic materials were synthesized by several hydrothermal carbonization methods, by which iron was provided in different ways: as FeCl_3_ prior to or during hydrothermal carbonization, as pure Fe particles, or as magnetic ferrofluid, followed or not by pyrolysis processes. The materials were thoughtfully characterized in terms of elemental composition, thermal degradation, porosity (N_2_ adsorption, SEM micrography), surface chemistry (FTIR spectroscopy, XRD diffraction), and magnetization curves on a self-made installation. The results indicated that the process design can significantly improve the structure and chemistry of the material, as well as the magnetization effect induced on the adsorbent. Fe as FeCl_3_ was more interesting in regards to the development of porosity, mainly creating micropores, although it did not provide magnetism to the material unless a further pyrolysis was applied. Thermal treatment at 600 °C did not only increase the BET-specific surface (S_BET_) (262 m^2^ g^−1^) of the hydrochar, but also involved the transformation of Fe into magnetite, providing magnetic behavior of the hydrochar. Increasing pyrolyisis temperature to 800 °C even enhanced a better development of porosity (S_BET_ of 424 m^2^ g^−1^) and also increased the specific magnetic susceptibility of the hydrochar as a result of the further transition of Fe into wustite and hydroxi-ferrite.

## 1. Introduction

During the last decade, many researchers all over the world working on carbon materials have looked at the process called hydrothermal carbonization (HTC) as a novel and promising greener route (as compared to other thermochemical processes) to transform biomass into carbon materials with greater energy content than the precursor, and/or with incipient porosity to be used in adsorption processes after diverse treatments, depending on the final purpose [1,2]. By HTC, the biomass is added to a water solution and the system is heated in an airtight recipient, so that an autogeneous pressure (corresponding to the saturation temperature) is generated. The result is the occurrence of a complex set of degradation reactions that eventually give rise to a solid phase (hydrochar, HC) with enhanced C content, a liquid phase (rich in abundant and diverse organic and inorganic compounds), and a negligible amount of gas (mainly CO_2_) [3].

The solid product obtained by HTC of lignocellulosic biomass is, in general, a low porosity carbon material, with most of its pores in the range of meso and macropores [4,5]. The degradation of compounds occurs as a result of hydrolysis, decarboxylation, decarbolynation, or dehydration, among others, with the produced compounds migrating to the liquid phase, which may be subsequently adsorbed on the HC surface or recombined by condensation and polymerization processes. A wide body of bibliography has stated that the recombination compounds may also produce a secondary char [6], evidenced, for example, by the formation of microspheres. The deposition of these compounds may inhibit the development of microporosity or their blockage, so HCs need further processing to develop their incipient porosity. Regarding the HC surface functionalities, most authors have reported a wide variety and abundance of surface groups, but a clear predominance of acid oxygen compounds; this is consistent with the HTC process, since, as biomass degrades, hydronium ions catalyze the reactions with a progressive acidification of the reaction media. 

The authors have previously explored how the porosity and surface chemistry of HCs can be tuned up to satisfy specific demands; for example, by using different gasifying agents during physical activation of HCs [7] or adding chemicals during HTC processes [8]. The optimization of both features is essential to guarantee selectivity towards a particular contaminant.

Providing magnetic properties to adsorbents can also be desirable from a practical point of view, in order to facilitate the separation of the adsorbent after being used. This is especially desirable in some applications, such as liquid adsorption processes [9] or in soil remediation processes where heavy metals have to be mobilized [10].

To the best of the authors knowledge, the research made on the production of porous magnetic biomass-based materials using HTC processes is very scarce. Zhu et al. have done remarkable work in this field [11,12,13]; these authors have studied the three-step post-treatment of HCs made from *Salix psammophila* following the dispersion of the HC on a FeCl_3_ solution, subsequent 700 °C pyrolysis, and further washing with HCl. They also tested the production of magnetic composites (MCs) by simultaneous ZnCl_2_ and Fe salt (as FeCl_3_) chemical activation of HCs previously prepared from sawdust and identified that iron salts may play a main role in the activation of the HC.

This work aimed to provide scientific insight about the production of magnetic porous materials from biomass following facile and sustainable HTC, with special emphasis on the simplification of the process stages. Almond shells, a very abundant biomass resource in Spain, were used as precursor and subjected to HTC processes following novel different methods by which Fe was added to the biomass, as described below. After the process, the carbon densification and reactivity of the process was evaluated, as well as the changes in the thermal degradation profiles of the HCs prepared. The mechanisms associated with the development of porosity on the HCs, which was only significant if postpyrolysis was applied, were discussed, and the changes in the forms of Fe during the subsequent stages was also studied. The magnetic susceptibility of the materials was also investigated by a self-made installation, under static conditions. The oxidation of Fe and its influence on both the opening of pores and supply of magnetic responses on the HC were analyzed.

## 2. Discussion of Results

### 2.1. Hydrocarbonization Processes. Reactivity and Thermal Behaviour

The solid yields obtained from HTC under the different conditions described in the experimental section are shown in Table 1. In relation to pristine precursor, HTC brought out a clear increase in the HHV of the materials, as a result of the expected carbon densification, which was clearly slighter in the case of the sample treated with FF. As it is well-known, the HTC process is initiated by dehydration and hydrolysis reactions, followed by decarboxylation and decarbonylation processes, which bring out the degradation of, first, the extractives and then the cellulosic fractions of the biomass, while lignin is mostly unaffected. Also, provided the time period is long enough, the degradation compounds can recombine by means of condensation, repolymerization, and aromatization processes, and create a lignite-type material that tends to be recombined with the HC. On the other hand, previous works have related the presence of lignin to the formation of a protective shell around the feedstock that might impede degradation to some extent [14].

Besides, the presence of Fe can catalyze the process or inhibit it, depending on the material and on the form under which Fe is found in the precursor. It is well-known that metal iron in the presence of water and oxygen and under high temperature tends to oxidize in several stages until iron oxide is formed on its surface. On the other hand, it is worthy to highlight the difference between the solid yields obtained by both methods (methods 1 and 2), suggesting that the content of FeCl_3_ retained on the HC is greater in the case of the second method.

Also, the presence of ferromagnetic fluid also involves a decrease in the solid yield of the HC (CA–HC–FF) as compared to standard procedures (CA–HC). The ferrofluid is composed of colloidal magnetic particles (such as magnetite, Fe_3_O_4_) diffused in an oily fluid; these particles, or the organic compounds belonging to the oils, may catalyze the degradation of the biomass.

Moreover, the greater ash content, in coherence with the Fe adsorbed on the HC, is responsible for the decrease in the HHV found in the Fe-catalyzed HCs. Further pyrolysis, as expected, yielded a greater carbon densification and, therefore, a bigger heating value, especially at 800 °C. The loss of volatile matter (with corresponding H and O removal), mainly in the form of cellulose and hemicellulose, is consistent with this carbon densification, which will be explored in details with the aid of DTG analyses.

The TGA and DTG curves corresponding to the HCs prepared under the conditions described above have been plotted in Figure 1 (a and b, respectively), while some comparative representative parameters have been included in Table 2. In particular, the temperature of the highest TG peak (the peak that is associated with hemicellulose and cellulose degradation), T_p_, was included, as well as the residual mass (%) measured at: (a) 350 °C (m_350_); (b) T_p_ (m_Tp_); and (c) final temperature, T_f_ (m_Tf_). Moreover, the main peak temperature range was identified for each sample and its corresponding activation energy (E_a_, kJ/mol) was determined, following an Arrhenious first order model.

It was found that the addition of Fe during HTC can have a slight or a significant effect on the thermal degradation profile of the HC, depending on the form under which it is added to the reaction system. In general, only the addition of FeCl_3_ had a remarkable effect on the TG profiles. This is outstanding, especially if one considers the solid yield values (SY, %) displayed in Table 1. In this table, it can be seen that using Fe under any of the forms studied involved a greater SY. This can be related, in the first place, to the enhanced ash content, as expected because of Fe deposition (that was further corroborated by SEM-EDX analyses) and/or to an inhibition of biomass degradation upon HTC/enhancement of recondensation and recombination reactions in the liquid phase, to form the “second char” or coke. TGA profiles showed that only FeCl_3_ gave rise to a more resistant material towards heating treatment (see values of final mass loss in Table 2), thus confirming that this compound effectively catalyzed the previous HTC.

On the contrary, the HC made with FF was more reactive at the first stages of deposition; in other words, the HC prepared with FF resulted in more vulnerability to heating at T < 350 °C (see in Table 2 how for CA–HC–FF, the remaining mass was lower at 350 °C (m_350_, %), as compared to the other HCs). At the end of the process, however, this sample showed the lowest residual mass FF; this is interesting since a greater devolatilization during HTC can be related to some extent to a less complete previous HTC. Regarding the HHV of this sample (the lowest), it could be inferred that, as a whole, this material underwent the lowest carbonization.

By applying a first order reaction model, an estimation of activation energies, at the highest degradation rate (around 400–500 °C, depending on the sample), allowed a more detailed understanding of reactivity of the HCs (results included in Table 2). Thus, CA–HC and CA–HC–Fe showed similar E_a_ values, whereas CA–HC–Fe_3_ had the highest activation energy. These results are in accordance with the DTG results.

### 2.2. Surface Characterization of Hydrochars

#### 2.2.1. N_2_ Adsorption Analyses

The porosity of the hydrochars was examined by means of N_2_ adsorption at −196 °C; the adsorption isotherms have been included in Figure 2, while typical parameters, as deduced from experimental adsorption data, are collected in Table 3.

The adsorption isotherms confirm the low pore volumes and mesoporous character of the HCs obtained by the standard method (CA–HC) as a result of pore blockage due to the permanence of reaction products, as usually found in the bibliography [8,15]. This porosity is mainly found in the mesopore range, as suggested from the low N_2_ adsorption at low relative pressures and further increase at high pressure values. The existence of hysteresis is also indicative of a structure mainly composed of big pores or cavities.

The presence of pure iron particles in the aqueous phase during HTC did not have a relevant impact on the pore size distribution of the HC (Figure 2a). In this case, the wider porosity of CA–HC–Fe may be due to the reaction of the biomass oxygen and the Fe, which may have enhanced the removal of amorphous carbon from the HC surface, as will also be corroborated by SEM imaging. In relation to the presence of FeCl_3_ in the liquid phase during the HTC, this compound involves a slight increase in the microporosity of the HCs as compared to the standard methods (Figure 2b). Moreover, the further pyrolysis of the HCs brought up an important development of microporosity; in fact, the shape of the isotherms changed from type III to type I, typical of microporous materials. As can be seen from Figure 2b, the porosity was located along the whole range of relative pressures. For 800 °C, S_BET_ values up to 467 m^2^ g^−1^ were achieved. Moreover, the isotherms did not present a defined knee at low P/P_0_ values, suggesting a wide microporosity distribution, and also a greater volume of mesopores at high P/P_0_ values. This value of S_BET_, as well as the clear microporosity unblockage and development, is similar to that obtained by other authors following Fe_3_Cl chemical activation of HCs [11]. Previous pieces of research have reported that during pyrolysis, the Fe^3+^ ions are hydrolyzed to amorphous Fe species (Fe(OH)_3_ y FeO(OH)) at temperatures lower than 350 °C. Subsequently, these species are converted into Fe_2_O_3_ below 400 °C; at greater temperatures (500–700 °C), the Fe_2_O_3_ should be reduced to Fe_3_O_4_ with the aid of reducing components such as amorphous carbon or gaseous CO. Finally, both Fe_2_O_3_ and Fe_3_O_4_ should be reduced to amorphous carbon, giving rise to metallic iron. According to our results, and in line with the research made by X. Zhu et al. [13], the formation of Fe oxides during pyrolysis is responsible for the formation of porosity in the HCs.

Finally, the presence of ferromagnetic fluid also had a negative impact on the porosity development of the HC (Figure 2a). This effect, probably due to pore blockage by the iron particles, was more evident for AS, probably die to its greater contribution of external surface, which may facilitate the accumulation of iron particles.

#### 2.2.2. Surface Morphology Analysis by SEM Imaging

The effects of the addition of metal compounds on the surface morphology of the HCs was followed by SEM micrography. For the sake of brevity, only those images showing outstanding features have been collected in this work, and are shown in Figure 3 and Figure 4.

Firstly, for the standard HC (CA–HC), typical previously reported features can be identified, that is to say, an external surface formed by big cavities and the presence of amorphous carbon particles, as well as microspheres [15]. This morphology did not change with the addition of Fe particles (CA–HC–Fe), although a slight decrease on the amorphous carbon matter can be suggested (cleaner surface), which may be attributed to the reaction enhanced by this metal that made this amorphous fraction more reactive.

Adding FeCl_3_ to the solution did not seem to involve a change on the surface morphology of the HC, although a decisive effect was found when the sample was subsequently subjected to pyrolysis at 600 and 800 °C (see Figure 3c, representative of sample CA–HC–Fe_3_-600). In the latter case, a great amount of dirt was removed from the HC surface and the tunnels appear connected, with pores in their walls at the same time that spheres were still preserved. The amorphous carbon was probably oxidized during the pyrolysis.

Incorporating EDX to the SEM analysis allowed us to identify how the Fe was distributed on the surface of the HCs. As an example, Figure 4 shows two of the images found for CA–HC–Fe_3_-600. By increasing the magnification of the figure, Fe particles of about 100 × 250 nm could be identified. Moreover, it could be observed that these particles were quite homogenously distributed on the HC surface. The pitted surface also suggests the development of micropores, as previously found by N_2_ adsorption.

The EDX analysis of the sample of CA–HC–Fe_3_-600 confirmed a Fe mass percentage of about 4% (slightly higher than that obtained by Zhu et al., described in ref. [13]), and C and O fractions of 84% and 12%. The presence of Cl (0.6%) suggests that the material washing was not enough and should be prolonged in further experiments.

#### 2.2.3. Surface Chemistry Analysis

The way in which Fe is added to the reaction media barely or significantly affected the surface chemistry of the corresponding HC, as it is explained next (Figure 5). For example, the presence of Fe particles (Figure 5a) did not have any effect on the number, location, or intensity of the spectral bands. Dissimilarly, the presence of FeCl_3_ (Figure 5b) in the solution involved clear modifications regarding the surface functionalities of the HC. In fact, the bands at 1000 and 1220 cm^−1^ were removed, indicating that FeCl_3_ enhanced the removal of a fraction of the cellulose of the HC, which is consistent with the increment of the acidity of the solution in the presence of the compound, and thus the promotion of a degradation effect [15]. Consistently, this sample showed a lower residual mass at 350 °C, as compared to the HC made without Fe (90.6% against 94.6%, as found in Table 2). After pyrolyisis, many acidic groups were removed (see absence of bands in the range 1200–1500 cm^−1^), at the same time that additional bands typically associated to chars (near 1600 cm^−1^, for example) corresponding to vibrations ν(C=C) in aromatic rings and ν(C–O) in alcohols, esters, or eters were observed.

On the other hand, it can also be corroborated that the addition of ferrofluid (compare CA–HC and CA–HC–FF in Figure 5c) did not involve a significant difference on the spectra and most of the surface groups remained the same. The appearance of a new intense band could be observed at approximately 580 cm^−1^, which might be attributed to the vibration ν(Fe–C) [16], and would confirm the presence of Fe_3_O_4_ on the HC.

### 2.3. Magnetization Properties

Due to the complexity of the installation for the measurement of magnetic properties, only those HCs reacting to a magnet were subjected to these tests. As expected, only the HCs that had been pyrolyzed after HTC were magnetic; the removal of HC oxygen surface groups during pyrolysis allowed their interaction with the HC Fe compounds and, therefore, their oxidation to forms such as magnetite or wustite, which was then corroborated by X-ray diffraction, as it is explained next.

Figure 6 shows, as an example, the behavior of CA–HC–Fe_3_ and CA–HC–Fe_3_-600. Other experiments in the presence of water showed that pyrolyzed particles could be easily attracted by an external magnetic field and the clear solution could be easily removed by a pipet.

The values of magnetic susceptibility (X) suggest that the materials were only slightly ferromagnetic, although they could be removed by a magnet at room temperatures. Best values were obtained for 800 °C (2.7, against 0.514 obtained for the sample made at 600 °C), which might be related to the improvement in pore volumes and the presence of wustite (20.6%) and magnesium ferrite (14.24%), although magnetite was destroyed, as confirmed by X-ray diffraction analyses. For the sample prepared at 600 °C, a surprisingly big proportion of magnetite was found (98%) in the crystalline part of the HC (39%), although, as a whole, the resulting material was less ferromagnetic. 

The crystallinity proportion allowed the recalculation of (X) and it was found that excluding the nonmagnetic part of the HC surface led to X values closer to those of ferrite (12.7).

Future works will investigate the use of different proportions of Fe during HTC and will improve the static experimental installation used for the magnetic analyses, opening these measurements to vibrating methods, which will provide more accurate results.

## 3. Experimental: Materials and Methods

### 3.1. Precursor: Almond Shell

Almond shell (AS) was chosen as precursor because of its convenient lignocellulosic composition with high lignin content (24.8%), hardness, homogeneity, and abundancy in the Extremadura region (Spain). The potentiality of this material as starting material towards HTC and activation processes, to give rise to porous materials and also biofuels, has been previously proven by the authors [17]. The biomass was supplied from Bioterra Pasat-Profuse (Badajoz, Southwest Spain). The materials were first washed thoroughly with water to remove dirt, and then dried in an oven, crushed, and sieved to a particle size of 1–2 mm.

Its proximate analyses, determined with a LECO analyzer CHNS (EA 1108), showed contents of C, N, H, O, and S of 51.3, 0.09, 5.81, 42.6, and 0.21%, respectively. The immediate analyses gave a typical composition of a lignocellulosic material (70.7% of volatile matter, 26.2% of fixed carbon, 2.3% of ash, and 1.1% moisture), similar to that previously found for other biomasses such as walnut shells, wood, or olive stones [18].

### 3.2. Methods

As it is described below, the effect of iron on the HTC process was studied using three different forms of this metal: (a) iron metal particles, (b) iron (III) chloride 6-hydrate, and (c) ferrofluid (EFH1). While a and b were purchased from Panreac Applichem (Barcelona, Spain), the ferrofluid (FF) was supplied by Ferrotech (Santa Clara, CA, USA). The incorporation of each chemical into the HTC process (and the process itself) was performed as it is explained below.

#### 3.2.1. Standard HTC Processes

The standard HTC processes were performed in a stainless steel autoclave (Berghof, Berlin, Germany). In a 0.2 L teflon vessel (unstirred), an appropriate amount of biomass precursor (3 g) and 50 mL of deionised water at room temperature were added. Then, the teflon vessel was sealed and placed into the autoclave and the system remained at room temperature overnight. Thereafter, the system was heated up in an electric furnace at 230 °C for 20 h (these reaction conditions were chosen based on previous HTC studies with other biomass sources using the same equipment) [15]. When the reaction time was reached, the autoclave was removed from the oven and subsequently placed in a cold-water bath and allowed to cool down up to room temperature. 

After cooling, the solid phase was separated from the liquid by vacuum filtration and subsequently dried at 80 °C to remove residual moisture. The dried HC was stored in closed flasks placed into a desiccator until further analysis. The experiments were carried out under autogeneous pressure in an autoclave, without the possibility of measuring interior conditions, but according to our studies, the pressure inside the vessel equaled that of the water at saturated conditions.

#### 3.2.2. Modified Hydrothermal Carbonization Processes

##### Processes Catalyzed with Pure Iron Metal Particles

In this case, 2 g of iron metal particles were mixed with 3 g of biomass and 50 mL of distilled water and the HTC process was then performed under standard conditions. Thereafter, the resulting HC was filtered as mentioned above and iron particles ware retired with the help of a magnet. The HC obtained was named CA–HC–Fe.

##### Processes Catalyzed with FeCl_3_

The addition of iron as FeCl_3_ to the HCs was performed following the methods proposed by Gai et al. [19], with some modifications, as described next. A Fe-containing solution (50 mL of FeCl_3_·6H_2_O with distilled water, 0.25 M) was added to the biomass (3 g) and directly subjected to HTC, performed under the conditions of standard experiments (Section 2.2.1). The name of this sample was CA–HC–Fe_3_.

The changes induced on the HC sample as a result of subsequent pyrolyisis were studied at two temperatures (600 °C and 800 °C, N_2_, 100 mL min^−1^). After pyrolysis, the samples were washed with HCl acid (0.1 M) and distilled water until they reached a neutral pH. Pyrolysis processes were done on a fixed bed cylindrical reactor inserted in an electric furnace, using an initial HC amount of 10 g. Detailed technical characteristics of the pyrolysis equipment have been described elsewhere [20]. In this case, the samples were named CA–HC–Fe_3_-600, and CA–HC–Fe_3_-800.

##### Processes Made in the Presence of Magnetic Ferrofluid

The procedure was analogous to the 2.2.2.1 method (2 g of FF was added to the system water biomass), although in this case it was not possible to remove the ferrofluid from the HCs. This HC was abbreviated as CA–HC–FF.

### 3.3. Characterization Techniques

#### 3.3.1. HC Reactivity and Thermal Behavior

Solid yield (SY, %) was calculated as the amount of solid product (i.e., HC) obtained after HTC, in relation to the initial mass of precursor.

Thermal analyses (TGA, DTG, and DTA) were performed using a SETARAM SETSYS EVOLUTION instrument controlled by a PC. Argon (100 mL min^−1^) was used as carrier agent and a heating rate of 10 °C/min was applied. Finally, the high heating values (HHVs) of the precursor and the HCs obtained were determined with a bomb calorimeter (Parr 1351).

#### 3.3.2. HC Porosity and Surface Chemistry

The porosity of the HCs was explored by N_2_ adsorption at −196 °C, using a semiautomatic adsorption unit (AUTOSORB-1, Quantachrome, Florida, USA). Prior to analyses, the samples were outgassed at 120 °C for 12 h. Experimental adsorption data were analyzed by suitable methods [21] to calculate: (a) the value of the BET-specific surface (S_BET_); (b) the external surface (S_EXT_), calculated by the α_s_ method, using the reference nonporous solid proposed by Carrott et al. [22]; (c) the percentage of internal surface (S_INT_), calculated as the difference between S_BET_ and S_EXT_; (d) the volume of micropores through the Dubinin–Radushkevich equation (V_miDR_); and (e) the volume of mesopores (V_me_), calculated as the difference between the pore volume at p/p_0_ = 0.95 and p/p_0_ = 0.10.

Also, the surface morphology of the samples was analyzed by scanning electron micrography (SEM, Hitachi, S-3600N, Krefeld, Germany) observation. The SEM samples were prepared by depositing about 50 mg of sample on an aluminum stud covered with conductive adhesive carbon tapes, and then coating with Rd–Pd for 1 min to prevent charging during observations. Imaging was done in the high vacuum mode at an accelerating voltage of 20 kV, using secondary electrons.

Finally, the surface chemistry of the HCs was evaluated by means of FTIR spectroscopy. FTIR spectra were recorded with a Perkin Elmer model Paragon 1000PC spectrophotometer (Waltham, Massachusetts, USA), using the KBr disc method, with a resolution of 4 cm^−1^ and 100 scans (Perkin–Elmer 1720, Waltham, Massachusetts, USA). The composition of the cristaline part of the HCs was also analyzed by X-ray diffraction, using Bruker equipment (Rhode Island, Warwick, USA).

#### 3.3.3. Experimental Set-Up for the Study of Magnetic Behavior

A self-made installation (Figure 7), as described below, was designed to measure the magnetic susceptibility of the HCs. A cylindrical plastic tube (0.005 m diameter) was filled with the HC sample and a copper wire was rolled around the tube (100 laps). The sample was then subjected to an external alternating magnetic field under different current intensities, and a pickup coil over the sample measured the electromotive force (EMF), which was then integrated to find the apparent magnetic flux and susceptibility.

Application of Faraday law allowed us to determine the value of the magnetic field from the EMF and HC bed section, and, from it, the magnetic susceptibility was calculated. A further comparative study with ferromagnetic and diamagnetic media will give an idea of the efficiency of the processes.

## 4. Conclusions

The physico-chemical form in which Fe is incorporated to the HTC process has a moderate effect on the process reactivity (slightly increasing the solid yield) and barely affects its thermal behavior and heating value (which decreases a little). In general, a mesoporous material with scant porosity covered by microspheres and Fe distributed quite homogeneously was found when Fe was added as metal particles, ferrofluid, or as FeCl_3_.

Adding a subsequent pyrolysis step did affect the pore development of the HCs and the carbon density, and also the surface chemistry of the materials, making them more interesting and stable for further applications.

Pyrolyisis at 600 and 800 °C can open the HC microporosity due to the existence of oxidation reactions that can play a dual role: a) removing the amorphous carbon of the carbon surface (as in the case of physical activation methods) and b) transforming the existing Fe into oxides that can provide magnetic behavior to the HC.

## Figures and Tables

**Figure 1 molecules-24-03996-f001:**
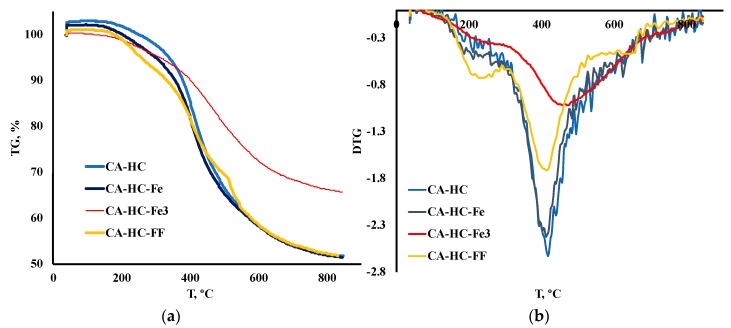
(**a**) TGA and (**b**) DTG degradation curves of HCs under inert conditions.

**Figure 2 molecules-24-03996-f002:**
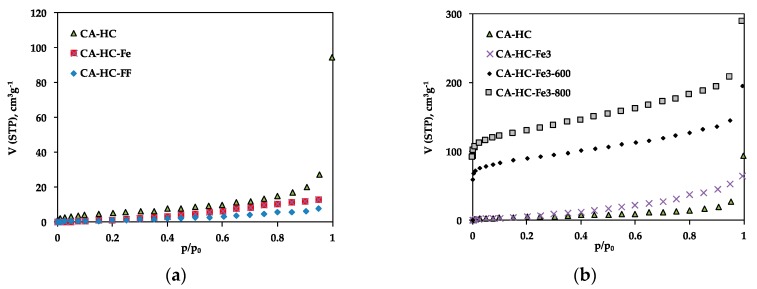
N_2_ adsorption isotherms at –196 °C of selected HCs: (**a**) CA–HC, CA–HC–Fe, and CA–HC–FF, and (**b**) CA–HC, CA–HC–Fe_3_, CA–HC–Fe_3_-600, and CA–HC–Fe_3_-800.

**Figure 3 molecules-24-03996-f003:**
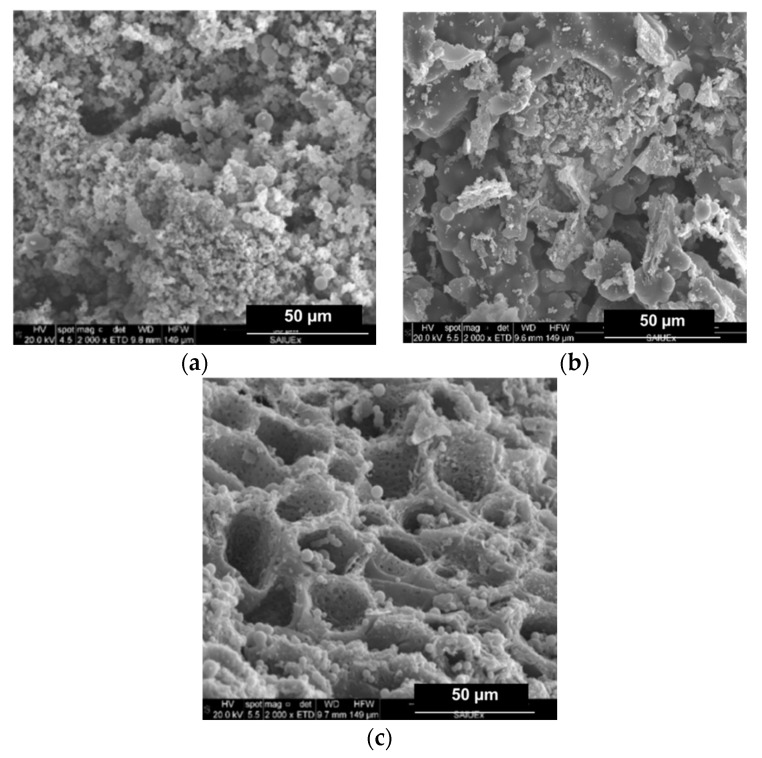
SEM micrographies of (**a**) CA–HC, (**b**) CA–HC–Fe, and (**c**) CA–HC–Fe_3_-600.

**Figure 4 molecules-24-03996-f004:**
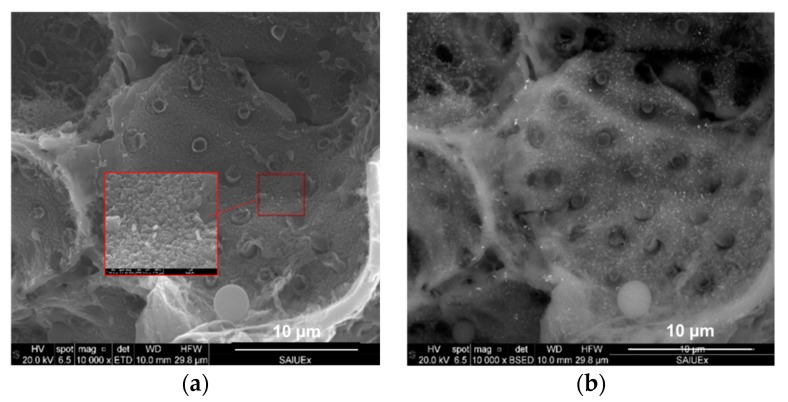
SEM micrographs incorporating (**a**) secondary electrons and (**b**) a detector of retrodispersed electrons.

**Figure 5 molecules-24-03996-f005:**
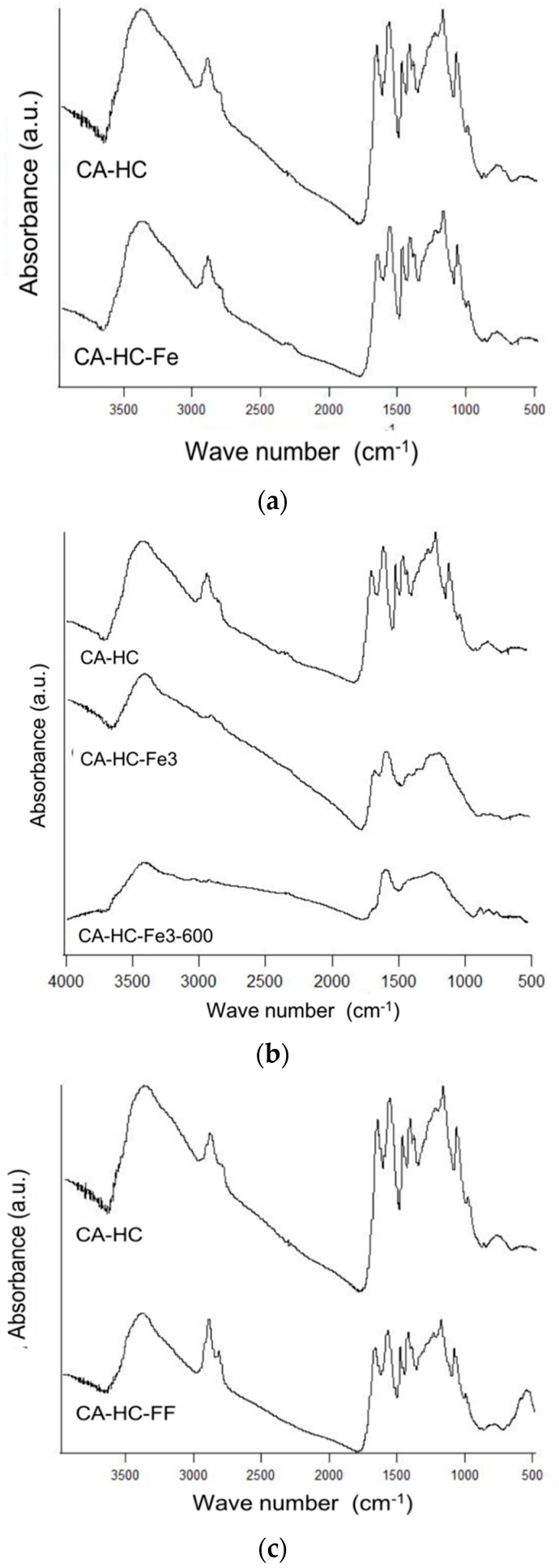
FT-IR spectra of CA–HC–Fe (**a**), CA–HC–Fe_3_ and CA–HC–Fe_3_-600 (**b**), and CA–HC–FF (**c**). The spectra of the sample CA–HC has been included in all cases to facilitate comparison.

**Figure 6 molecules-24-03996-f006:**
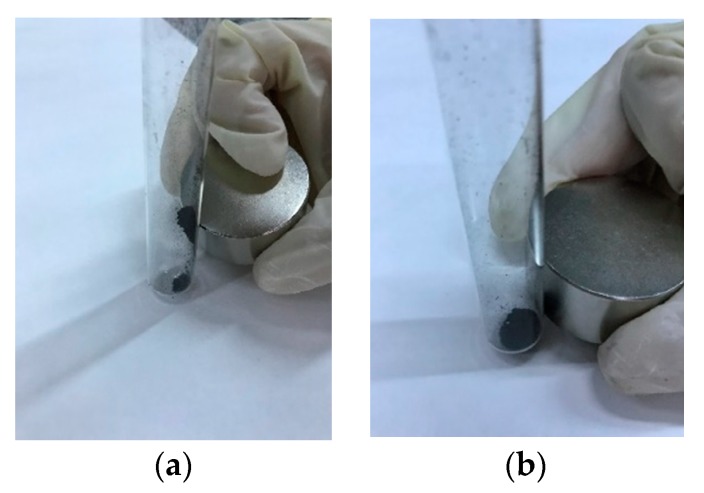
Representative behavior of (**a**) pyrolyzed ones (CA–HC–Fe_3_-600) and (**b**) nonpyrolyzed HCs (CA–HC–Fe_3_)**.**

**Figure 7 molecules-24-03996-f007:**
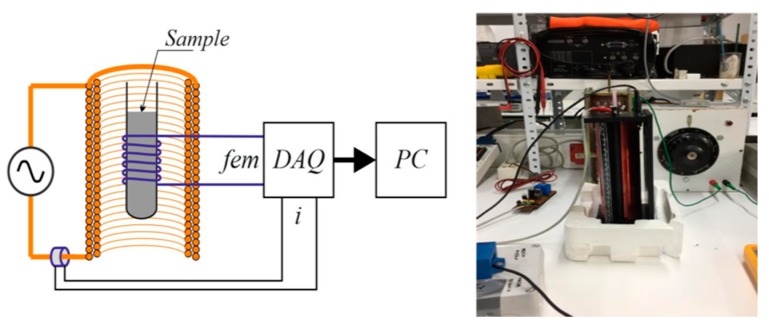
Self-made experimental installation for the measurement of magnetic properties.

**Table 1 molecules-24-03996-t001:** Values of solid yield (%) and high heating value (MJ Kg^−1^) of hydrochars (HCs).

Hydrochar	Solid Yield (%)	HHV (MJ/Kg)
**CA**	--	16.4
**CA-HC**	41	28.3
**CA-HC-Fe**	36	26.6
**CA–HC–Fe_3_**	30	23.3
**CA–HC–FF**	31.1	17.6
**CA–HC–Fe_3_-600**	15.2	27.6
**CA–HC–Fe_3_-800**	11.3	28.4

**Table 2 molecules-24-03996-t002:** TGA main peak temperature (T_p_, °C), representative mass loss values (%) at 350 °C (m_350_), and characteristic kinetic parameters.

	T_p_ (°C)	m_350_ (%)	m_Tp_ (%)	m_Tf_ (%)	E_a_ (kJ/mol)	T° Range (for Kinetic Study)
**CA–HC**	418	94.6	81.5	59.9	28.9	400–450
**CA–HC–Fe**	413	90.6	79.4	51.4	28.4	400–450
**CA–HC–Fe_3_**	456	93.1	84.6	65.7	35.1	450–500
**CA–HC–FF**	416	88.4	78.9	50.9	30.5	400–450

**Table 3 molecules-24-03996-t003:** Textural parameters of the HCs as deduced from N_2_ adsorption at −196 °C.

Hydrochar	S_BET_, m^2^ g^−1^	V_mi_, cm^3^ g^−1^	V_me_, cm^3^ g^−1^	V_T_, cm^3^ g^−1^	S_ext_, m^2^ g^−1^	S_int_, m^2^ g^−1^
CA–HC	20	0.009	0.033	0.042	29	-
CA–HC–Fe	12	0.004	0.018	0.022	22	-
CA–HC–FF	5	0.002	0.010	0.012	12	-
CA–HC–Fe3	36	0.012	0.070	0.082	87	-
CA–HC–Fe3-600	262	0.152	0.081	0.244	90	149
CA–HC–Fe3-800	437	0.199	0.123	0.321	195	229

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
