# Peer review of "Magnetic Behavior of Carbon Materials Made from Biomass by Fe-Assisted Hydrothermal Carbonization"

_molecules, 2019, doi:10.3390/molecules24213996_

Round 1

Reviewer 1 Report

This is a very interesting initial study in the area that has not been extensively explored so far and there is a lack of available data in the literature.

Paper seems a bit chaotic in some parts. However, it is possible to fix it - attached manuscript in pdf file has some more comments, that will hopefully help improving the paper.

Moreover, introduction seems a bit vague and can be improved by giving more comprehensive overview of the topic to the readers.

Author Response

Dear editor, 

attached I have included the answers to the reviewer, where I explain the changes we have made in the manuscript following his/her suggestions.

thank you very much

Reviewer 2 Report

The paper is well written and well structured. The title clearly describes the contents of the paper. The abstract provides a concise and complete summary and the reference list is appropriate. I would suggest the article for publication. Before final publication, a few comments that the authors could take into account:

Pg 2 Ln 83: “… was made as it is explained bellow” It should be “below”.

Pg 3, Ln 109, “the name of this sample was CA-HC-Fe“, I believe the sample is “CA-HC-Fe3”. Please double check.

Fig 1 (b) is not of professional quality, and doesn’t add any values to this manuscript. I would suggest using a conceptual scheme of the setup to have a clear visualization to support the text.

Figure 2: Why are Figure 2(a) and (b) identical, the only difference is the marker type of CA-HC-Fe?

Once the authors have decided that an abbreviation is used for a sample type, please use the abbreviation consistently thereafter for the general readability of the manuscript. For example, Table 3, is “CA-HC-Fe3-1” the same as “CA-HC-Fe3”? Figure 5: Is “CA-HC-Fe3-1-600” the same as “CA-HC-Fe3-600”?

Figure 3: multi-panel figures should be clearly labeled, such as (a), (b), (c). please also consider adding small gap lines between image (a) and image (b) to make them more distinguishable to each other. Scale bars are not readable.  

Figure 4: Scale bars cannot be seen, please consider making them larger and more readable.

Pg 13, Ln 1 and 2 come from nowhere, please revise. 

Author Response

ANSWER TO REVIEWER 2

First of all, we would like to thank the reviewer for his/her time and effort to improve our manuscript. It is really encouraging and motivating for us. Below, we explain the changes we have made, addressing the suggestions of the reviewer.

According to his/her suggestions, we have tried to improve the descriptions of the experimental section (adding more info about the precursor, about the equipment used) and also have enlarged the discussion at some points that we understand had to be more clear (here, we have corrected Figure 3 (previous figure 2) that was accidentally duplicated in the previous version of the manuscript and now contains the isotherms of all HCs.

We have also included more background at the introduction section and included new references, and we have carefully revised the whole manuscript, looking for typos and other mistakes, and of course including the suggestions made by the reviewer, including the modifications of Figures (we have modified the scale bars in SEM images and also the legend in the FT-IR graphs). We hope the new version of the manuscripts fulfils the requirements to be accepted at Molecules.

All changes have been marked in red color in the new version of the manuscript.

Reviewer 3 Report

The authors have put forward a nice paper wherein they used biomass to convert into carbon materials using hydrothermal carbonization. They have employed several analytical techniques to characterize the materials such as FTIR, N2 adsorption analyses, SEM, Thermal analyses and magnetic behavior. I would accept this article for publication once they address the following comments.

Please proofread the draft before the final submission and fix the English sentences by using proper grammar, typos and also when you use an abbreviation for the first time for eg. HC in the abstract, define it right there. The authors have defined later in the introduction. Please fix the chemical formulas such as FeCl3 instead of FeCl3. Notice the subscript. The reactions on page 7, line 250 and 251 are not balanced reactions. Please complete them. It is not clear why the authors have provided 2 figures in Figure 2 which are almost identical. No further explanation was provided on these 2 figures. If they are different, then, it would be better if they could name them as 2a and 2b. The authors need to provide a good explanation why the biomass materials have advantage compared to traditional porous materials. Also, there is no proper reasoning has been provided how/why the magnetic properties are useful. The authors could provide better explanation on the trend of surface areas of various HTC materials.

Author Response

ANSWER TO REVIEWER 3

We are very grateful to reviewer 3. The time and effort you have spent in order to improve our manuscript is very valuable for us; thank you very much.

In the new version of the manuscript we have made a effort to improve the writing of the whole paper, and have especially enlarged the introduction and discussion sections. In the first one, we have added more information about the concept of hydrothermal carbonization and the production of tunable porous materials from biomass. We have also improved the discussion and clarified better the trends and results obtained; we have also worked on the Figures, improving their legends and quality, and we have moved previous Figure S1 to the main document.

We hope the new version of the manuscript fulfills the requirements for its publication in the journal Molecules.